# Phylogenetics, Epidemiology and Temporal Patterns of Dengue Virus in Araraquara, São Paulo State

**DOI:** 10.3390/v16020274

**Published:** 2024-02-09

**Authors:** Caio Santos de Souza, Giovana Santos Caleiro, Ingra Morales Claro, Jaqueline Goes de Jesus, Thaís Moura Coletti, Camila Alves Maia da Silva, Ângela Aparecida Costa, Marta Inenami, Andreia C. Ribeiro, Alvina Clara Felix, Anderson Vicente de Paula, Walter M. Figueiredo, Expedito José de Albuquerque Luna, Ester C. Sabino, Camila M. Romano

**Affiliations:** 1Laboratório de Virologia, Instituto de Medicina Tropical, Universidade de São Paulo, São Paulo 05403-000, SP, Brazil; caszx4@usp.br (C.S.d.S.); giovanacaleiro@gmail.com (G.S.C.); clarafelixx@gmail.com (A.C.F.); anddeblair@yahoo.com.br (A.V.d.P.); 2Laboratório de Virologia, Instituto de Ciências Biomédicas, Universidade de São Paulo, São Paulo 05403-000, SP, Brazil; 3Laboratório de Parasitologia, Instituto de Medicina Tropical, Universidade de São Paulo, São Paulo 05403-000, SP, Brazil; ingra27@gmail.com (I.M.C.); jaquelinegoesdejesus@gmail.com (J.G.d.J.); thaiscoletti@gmail.com (T.M.C.); alves.camaia@gmail.com (C.A.M.d.S.); sabinoec@gmail.com (E.C.S.); 4MRC Center for Global Infectious Disease Analysis, Imperial College London, London SW7 2AZ, UK; 5Instituto Oswaldo Cruz, Salvador 21040-900, BA, Brazil; 6Serviço Especial de Saúde de Araraquara-SESA, Faculdade de Saúde Pública da USP, São Paulo 01246-904, SP, Brazil; aapcosta@sc.usp.br (Â.A.C.); minenami@gmail.com (M.I.); andreia_843@hotmail.com (A.C.R.); waltermf@sc.usp.br (W.M.F.); 7Departamento de Medicina Preventiva/Instituto de Medicina Tropical, Faculdade de Medicina, Universidade de São Paulo, São Paulo 05403-000, SP, Brazil; eluna@usp.br; 8Hospital das Clínicas da Faculdade de Medicina da Universidade de São Paulo (HCFMUSP), São Paulo 05403-010, SP, Brazil

**Keywords:** dengue virus, DENV-1, clade replacement, phylogeny

## Abstract

Dengue virus (DENV) is a prominent arbovirus with global spread, causing approximately 390 million infections each year. In Brazil, yearly epidemics follow a well-documented pattern of serotype replacement every three to four years on average. Araraquara, located in the state of São Paulo, has faced significant impacts from DENV epidemics since the emergence of DENV-1 in 2010. The municipality then transitioned from low to moderate endemicity in less than 10 years. Yet, there remains an insufficient understanding of virus circulation dynamics, particularly concerning DENV-1, in the region, as well as the genetic characteristics of the virus. To address this, we sequenced 37 complete or partial DENV-1 genomes sampled from 2015 to 2022 in Araraquara. Then, using also Brazilian and worldwide DENV-1 sequences we reconstructed the evolutionary history of DENV-1 in Araraquara and estimated the time to the most recent common ancestor (tMRCA) for serotype 1, for genotype V and its main lineages. Within the last ten years, there have been at least three introductions of genotype V in Araraquara, distributed in two main lineages (L Ia and L Ib, and L II). The tMRCA for the first sampled lineage (2015/2016 epidemics) was approximately 15 years ago (in 2008). Crucially, our analysis challenges existing assumptions regarding the emergence time of the DENV-1 genotypes, suggesting that genotype V might have diverged more recently than previously described. The presence of the two lineages of genotype V in the municipality might have contributed to the extended persistence of DENV-1 in the region.

## 1. Introduction

According to the World Health Organization (WHO), Dengue virus (DENV) stands as the most relevant arbovirus in the world, posing a threat to 3.9 billion people in 120 countries [1]. Annually, over 390 million infections are estimated to occur, with tropical and subtropical areas being the most affected ones, particularly in urban and developing regions such as countries in South America and Asia [1]. The International Committee of Taxonomy of Viruses (ICTV) classifies DENV within the *Flaviviridae* family, *Orthoflavivirus* genus, and it is currently known as the *Orthoflavivirus dengue* species [2]. Like other members of the *Orthoflavivirus* genus, the DENV genome is a single-strand, positive-sense, ~11 kb long RNA, which encodes three structural proteins (which are known as Pre-membrane/membrane, Capsid and Envelope) and seven non-structural (NS) proteins (NS1, NS2A, NS2B, NS3, NS4A, NS4B and NS5). DENV is classified into four serotypes (DENV-1 to DENV-4) due to antigenic and genetic differences, with each serotype further exhibiting various genotypes [3]. DENV serotypes can be defined as viruses that present 65% similarity in the amino acid sequence, while genotypes refer to groups of DENV that diverge no more than 6% in the nucleotide sequence [3]. The four DENV serotypes entered the human population at distinct moments, with DENV-2 being the oldest [4]. Specifically, DENV-1 presents five distinct genotypes, which coalesced at distinct moments. For example, genotype V is the one that predominates in Brazil, and it was believed that this genotype originated in 1935 [5]. 

Infection with any serotype can lead to dengue fever, a disease whose clinical manifestation includes flu-like symptoms such as fever, nausea, vomiting, rashes and pain (more specifically headaches and retro-orbital pain). It can progress to dengue with warning signs, which usually manifests 2 days after the fever ceases, and it is characterized by the presence of blood in vomit or stool, abdominal pain and bleeding from mucous membranes, such as the nose and gums, and ultimately progresses to a life-threatening condition known as severe dengue, which is manifested by the presence of shock, internal bleeding and respiratory distress, which can lead to death [1]. It is also estimated that one out of four infections are symptomatic [1]. In Brazil, from 2013 to 2022, a substantial number of possible dengue cases amounting to 9,938,730, with 5.836 deaths, were reported [6,7]. During this period, there was a clear pattern of alternating circulating serotypes between DENV-1 and DENV-2 in the country, except for the year 2013, when DENV-4 was the main circulating serotype [6,7]. In 2023, DENV-3 re-emerged in the country, with three autochthonous cases detected in the State of Roraima and one imported case detected in Paraná [8]. It is also well established that the clade replacement phenomenon, which is defined as the alternance among DENV serotypes, genotypes and lineages in a given location, plays an important role in dengue fever severity and viral maintenance [6]. The impact of DENV clade replacement on the number of dengue cases and severity was described in many different regions, such as Brazil, Panamá, Vietnam and Thailand [9,10]. 

In this scenario, the municipality of Araraquara emerged as a significant contributor to the escalating number of dengue cases [11]. Up until 2005, only sporadic cases were reported in the municipality. However, the panorama of dengue fever underwent a significant change in 2006, when serotype 3 was introduced into the region, leading to the first reported outbreak [11,12]. Subsequently, in 2010, the introduction of serotypes 1 and 2 resulted in a significant rise in the number of cases and deaths [11]. A similar situation was reported in 2023 by Gularte JS et al., 2023. In this study, it was reported that more than 67,000 cases of dengue fever in Rio Grande do Sul State were registered in 2022, and genomic analyses performed by Gularte and colleagues confirmed the predominance of DENV-1 genotype V in the region [13]. 

Particularly noteworthy were the years 2015, 2019 and 2022 in Araraquara, which reported 8209, 23,538 and 21,070 cases, respectively [14]. During the period of 2014–2018, DENV-1 was the main serotype circulating in the municipality, followed by clade replacement in 2019, when DENV-2 triggered a massive outbreak [15]. In this study, we performed genome sequencing analysis of DENV using samples collected from the municipality of Araraquara in 2015, 2016, 2019, 2021 and 2022 to assess the molecular characteristics of the circulating viral strains during this period.

## 2. Materials and Methods

### 2.1. Ethical Statement

This study was approved by the ethical committee of the Medicine School of the University of São Paulo, Approval under No. 4.519.364, encompassing authorization for sample collection, viral detection and sequencing of viral genomes.

### 2.2. Study Population

The study population is part of a cohort of children and youth (2–16 years old) that has been under observation since 2014 from Araraquara. Briefly, this cohort was built to assess the immunological status of the youth population in the municipality, as well as to evaluate, annually, the number of dengue cases in the studied group. The cohort consisted of 3514 individuals, who had serum collected at the time of entry in the study. After initial sample collection for serological status assessment, blood samples were collected again and sent to the Institute of Tropical Medicina (IMT-USP) if symptoms befitting dengue fever were reported by the family of the participants (named “fever case” samples). The complete cohort study is better described elsewhere [15]. These samples were tested by qPCR (described in following subtopic) and/or NS1 antigen and serology and from 2014 to 2018, and a total of 257 qPCR DENV-1 cases were confirmed. In the present study, we randomly included 50 samples representative of the positivity detected by year.

### 2.3. DENV RNA Detection and Sample Selection for Next-Generation Sequencing (NGS)

Viral RNA was extracted from 140 µL of plasma and/or serum using the QIAamp Viral RNA MiniKit (QIAGEN, Hilden, Germany) according to the manufacturer’s instructions, obtaining a total of 60 µL of RNA. For the initial detection of viral RNA, a probe-based qPCR capable of detecting the 4 DENV serotypes was performed [16,17]. Following the confirmation of DENV infection, samples were typed through specific qPCR [17,18]. DENV-1 samples with a cycle threshold (CT) < 30 were eligible for the Next-Generation Sequencing (NGS).

### 2.4. DENV-1 Complete Genome Amplification

The primers to amplify the complete genome of DENV-1 were designed using the online tool PrimalScheme (available at primalscheme.com) and a polifasta file containing reference DENV-1 complete genomes as input. After defining the desired amplicon size of 400 base pairs and an overlap of 75 nucleotides, the tool generated two sets of primers suitable for implementation in a multiplex scheme, arranged into two pools. These primer sets enabled the amplification of the complete viral genome [19]. 

Following the cDNA synthesis utilizing First Strand cDNA ProtoScript II (New England Biolabs, Oxford, UK), the two multiplex PCR reactions were conducted under the conditions described elsewhere [20]. Amplicons were purified using 1× AMPure beads (Beckman Coulter, High Wycombe, UK) and quantified using the Qubit 4 fluorometer (Invitrogen, Carlsbad, CA, USA). A DNA concentration >2 ng DNA/µL per sample was considered suitable for whole-genome sequencing. 

The libraries were generated using the genomic DNA sequencing kit SQK-LSK108, where 250 ng of barcoded PCR products were pooled in equimolar proportions. The libraries were loaded onto an Oxford Nanopore Flow Cell R9.4 and the data were collected after ~30 h [20].

### 2.5. Bioinformatics Workflow and Consensus Generation

The base-calling of the FAST5 reads containing the raw signal data was conducted using Guppy base-caller integrated within MinKNOW (https://community.nanoporetech.com/ (accessed on 10 September 2023). Then, FASTQ files were used to map the reads against a reference using the genome ID#KP188547 as the reference in the CLC genomics workbench (Qiagen, Hilden, Germany) [21]. Genomes with a coverage of >10 per base were exported in FASTA format for further analyses.

### 2.6. Envelope (E) Gene Sequencing 

The DENV genomes that were not suitable for Next-Generation Sequencing (NGS) underwent partial envelope gene sequencing using the Sanger method. We opted for this approach to incorporate the highest possible number of samples from Araraquara into the analysis. To amplify ~800 pb of the envelope, we performed PCR using the primers and protocol described by Bruycker-Nogueira F et al., 2015, with some modifications [22]. Briefly, the PCR was performed using 14.75 µL of Ultrapure Nuclease-Free Water (NFW), 1× PCR Buffer, MgCl_2_ 1.5 mM, 0.25 uM of each primer, 0.4 µM of dNTP and 1U of PlatinumTaq Polymerase in a final volume of 25 µL. 

Amplicons were sequenced using BigDye Terminator v3.1 on the ABI 3730 DNA analyzer (Thermo Fisher Scientific, Waltham, MA, USA). Electropherograms were analyzed and the consensus sequences were assembled using the Geneious Prime software v.2023.2.1 (Dotmatics, Hertfordshire, UK).

### 2.7. Multiple Sequence Alignment and ML Phylogenetic Reconstruction

The dataset comprised 284 partial E gene sequences (~800 nucleotides long) retrieved from GenBank (sequence list available in the Appendix A) containing sampling date, host and country information and 37 sequences generated in this study (either by NGS or Sanger). Sequences were multiple aligned using MAFFT v7.407 and visually inspected using the SeaView v5.0.4 software [23,24]. To first classify DENV-1 sequences into genotypes, we reconstructed a Maximum Likelihood phylogenetic tree using the IQ-TREE 2 under the GTR + I + G nucleotide substitution model, as it was the best-fitting one selected by ModelFinder implemented in IQTREE [25]. The phylogenetic tree was visualized with FigTree v.1.4.4 [26].

### 2.8. Coalescent Analysis and Molecular Clock

The time to the most recent common ancestor (tMRCA) of the DENV-1 genotypes, and particularly of the Araraquara ones, was estimated using the Bayesian Markov Chain Monte Carlo implemented in BEAST 1.10.4 [27,28]. The Bayesian Skyline (BSL) coalescent method was performed with a relaxed uncorrelated exponential molecular clock using time-stamped data scaled in years using the GTR + I + G nucleotide substitution model. The convergence of parameters was inspected with Tracer v.1.7.2 with uncertainties addressed as 95% highest probability density (HPD) intervals. 

After 100 million runs, the trees sampled at every 10,000 steps were summarized in a maximum clade credibility (MCC) tree with 10% burn-in using TreeAnnotator and visualized with FigTree v.1.4.4. In addition to the BSL method, the exponential growth tree prior was run for comparison.

## 3. Results 

A total of 37 samples were positive on the qPCR for DENV-1 with CT < 30, median 24.5 (18.4–30.7), and were submitted to the whole-genome amplification. The number of samples per year was as follows: 2015 (*n* = 26, collected from March to May), 2016 (*n* = 1, collected in April), 2019 (*n* = 2, from February), 2021 (*n* = 2, collected in June) and 2022 (*n* = 6, collected in May). Twenty-eight out of the thirty-seven samples submitted to the whole-genome amplification had a sufficient DNA concentration and were sequenced using the Nanopore method. The remaining samples (*n* = 9) were submitted to Sanger sequencing (envelope gene). Maximum likelihood analysis of DENV-1 showed a clear separation of the distinct genotypes and the genotype V division into lineage I and lineage II, previously classified by Bruycker-Nogueira and colleagues [9] (results not shown). The genomes sequenced in this study can be found in the Genbank public database under the accession numbers OR553381 → OR553391, OR545518 → OR545534 and OR520604.

We then ran the same dataset in BEAST to perform the temporal estimates. According to the BLS estimates, DENV-1 coalesced approximately 138 years ago (95% HPD 102–188 ya) in 1884, while the origin of the genotype V was estimated to be around 57 years ago (95% HPD 50–67 ya) in 1965. Similar data were found under the exponential growth.

The MCC tree also showed the two different lineages of genotype V [9]. The genomes from 2015 and 2016 were classified as lineage II (LII), while genomes collected from 2019 onwards were classified as lineage I (LI). According to Bayesian estimates, lineage II coalesced in 1982 (around 41 years ago), lineage I coalesced in early 1977 (around 46 years ago) and lineages Ia and Ib both coalesced in 2001 (Figure 1).

All but five viruses from Araraquara 2015/2016 clustered together in a branch that probably emerged in 2009 (95% HPD 2006–2011). Two out of the remaining samples from 2015 clustered with samples from Central (2013) and Southeast (2013) regions. This second cluster consists primarily of Brazilian DENV strains, comprising viruses from Goiás and Rio de Janeiro (2013), Ceará and Sergipe (2014) and São Paulo (2016). Notably, these two clusters constitute the entire lineage II. Viruses from Araraquara, 2019, belong to lineage Ib and are grouped with other viruses from Southeast Brazil (2016 and 2018). 

The cluster also includes viruses from the North (2010) and Northeast (2014) regions of the country. The tMRCA for this group was estimated at 16.5 years ago, in 2006 (95% HPD 2004–2008). It is also worth noting that, in contrast to lineage II, some of these genomes were closer to other South American countries, such as Venezuela and Colombia. The DENV-1 sequences obtained in 2021 and 2022 clustered together and were close to other genomes from the Southeast (2016, 2021, 2022 and 2023) and Central regions (2016) of the country. The node comprising these sequences is dated to 9.5 years ago (95% HPD 8–10 ya), in 2013.

Finally, the dates of key DENV-1 events were re-estimated. According to our results, DENV-1 had its origin 138 years ago, estimated in 1884 (95% HPD 1833–1920), and genotype V arose 58 years ago, in 1965 (95% HPD 1955–1972). The approximated origin of genotype V lineages and sublineages as well as the tMRCA of the other four genotypes is described in Table 1.

## 4. Discussion

The phylogenetic structure of DENV in endemic regions is marked by lineage turnovers, which cause significant impacts on the epidemiology of dengue due to the susceptibilities to cross-reactive immune responses against the currently circulating virus. Brazil annually reports the highest number of dengue fever cases and deaths, and clade replacement is believed to play a great part in these numbers; therefore, constant vigilance for the introduction of new clades in endemic regions is of utmost importance [6,9,10,29,30,31,32,33]. 

Araraquara, located in the central region of São Paulo state, is among the most industrialized cities in the state and reported the first autochthonous dengue case in 1996. However, information regarding the serotype was not available until 2006, when DENV-3 was described in the region. Since then, the number of cases progressively rose, leading to severe outbreaks and epidemics every year [9,11]. After the emergence of DENV-1 in 2010, Araraquara experienced an extended predominance of this serotype for several years with a remarkable number of reported cases over the years [11]. Particularly, from 2014 to 2018, DENV-1 emerged as the predominant serotype in Araraquara, while DENV-2 dominated from late 2018 to 2019, leading to more than 20 thousand reported cases [11,14,15]. 

Nonetheless, in 2020, DENV-1 reclaimed its status as the prevailing serotype and has continued to do so up to the present time [34]. The phylogenetic analysis using longitudinal DENV-1 samples collected in an 8-year time span performed in this study represents the first description of DENV-1 lineages circulating in the municipality of Araraquara. During this period, we detected the presence of the two lineages of DENV-1 genotype V in Araraquara, L I and L II [29]. Initially, DENV-1 was represented exclusively by lineage II and almost all viruses clustered together. Only three samples (two from 2015 and one from 2016) did not cluster with the remaining viruses collected in 2015, though we cannot definitively determine whether this is due to a distinct introduction of a very similar strain or a result of locally accumulated diversity. From 2019 onwards, viruses from L II were no longer sampled, and all circulating viruses until 2022 belonged to lineage I. Particularly, viruses from 2019 clustered within sub-clade L Ib, while those from 2021 and 2022 belonged to the sub-clade L Ia, clearly revealing a second clade replacement. Indeed, the most recent spread of lineage I across the country is also supported by a study by Gularte et al., 2023, in which the phylogenetic analyses of 70 DENV-1 genomes from 2022 clustered within lineage I [13]. 

The presence of distinct lineages and sublineages of DENV-1 in Araraquara might have played a critical role in sustaining the prevalence of this serotype in the region for such an extended period, as it has been seen elsewhere [32]. Unfortunately, we do not have samples prior to 2015 and also no samples were collected in 2017 and 2018, so it is impossible to precisely determine how many clade/lineage replacements occurred since DENV-1’s emergence in Araraquara. Nevertheless, the continual emergence of distinct lineages might be the key to the observed time frame of DENV-1 predominance. The isolates detected in 2019 (lineage Ib) were not able to spread, maybe due to the dominance of DENV-2 at that time. But two years later, the number of DENV-1-susceptible hosts was sufficient to allow the DENV-1 lineage Ia to spread and replace DENV-2, causing again a major outbreak.

The phylogeny also evidenced two distinct clusters within lineage I, sufficiently divergent from the previously described L Ia and L Ib [9], which could be named L Ic and Id here, but the low posterior probability did not allow us to properly propose these classifications. Although these clusters also comprise Brazilian viruses, no sample from Araraquara was classified as these new sub-lineages. 

While several studies point out that clade replacement is a stochastic event and does not depend on selection, there is also evidence that strains of DENV can exhibit different fitnesses either in mosquitoes or in vertebrate hosts [10,35,36,37]. For instance, during a clade replacement in Indonesia within the Cosmopolitan genotype of DENV-2, the increased number of cases was associated with a higher fitness of the new clade in mammalian cells [38]. Another relevant topic of clade replacement was demonstrated in a study by Katzelnick et al., 2021. It was shown that within one serotype, genotypes can also present significant divergence at the antigenic level, with one genotype being more antigenically related to sequences from other serotypes than to sequences from the same serotype. For instance, DENV-3 genotype II is much more similar to other serotypes than to DENV-3 genotype III. These findings support the hypothesis that within one serotype, genetic and antigenic divergences play important roles in serotype maintenance and number of cases/severe cases [39].

Our group recently reported a clade replacement within the DENV-2 American/Asian genotype (III) in São Paulo, where the introduction of the BR-4 lineage was associated with major outbreaks in 2019 and rapid expansion of the lineage in the country [31]. Although we have not evaluated the number of deaths and severe dengue cases in the 2015 and 2022 DENV-1 epidemics, the higher number of cases caused by lineage I compared to lineage II was noticeable (i.e., 8209 in 2015 and 21,070 reported cases in 2022).

Our Bayesian analysis also enabled us to re-estimate the dates of key DENV-1 events. According to the estimates, the origins of serotype 1 as well as the genotypes I to IV were similar to those reported elsewhere [5]. However, we noticed that the clustering pattern of two sequences (AF425620, from Cote d’Ivoire and AF425625, from Nigeria), previously classified as genotype V [40], were not consistent and grouped near genotype III in the ML (not shown) and MCC trees. The previous reports that considered these sequences to be genotype V dated the origin of this genotype back to 1935 [34,35]. However, according to our estimation, genotype V might have emerged much more recently, around 1968. It has to be noted that both previous studies included a limited number of genotype III reference sequences [5,40]. Consequently, the lower genetic diversity of genotypes III and V from the previous phylogenetic analysis might have hindered accurate classification during the reconstructions. In an effort to explore the potential factors underlying the discrepancies observed in previous phylogenetic classifications, we tested these samples for recombination and estimated the genetic distances between them and others from genotypes III and V. RDP [41] found no evidence of recombination in both sequences. Genetic distance analysis using the Kimura 2 parameter revealed >6% distance between these genomes and those from genotype V, but a 5.2% distance from viruses from genotype III. 

Based on previous estimates that classify DENV as distinct genotypes when the genomes differ by around 6–8% at the nucleotide level [3,42], one might suggest that these viruses might be more appropriately classified as genotype III. Finally, the Dengue genotyping tool (available at https://www.genomedetective.com/app/typingtool/dengue/ (accessed on 12 September 2023) was unable to classify AF425625 at the genotype level, but AF425620 was classified as “related but not part of DENV-1 Genotype III”. 

We did not go further on the analysis of these two genomes but it is likely they represent either atypical genotype III viruses or sylvatic strains, so we opted to include them as genotype III in our tMRCA estimates.

## 5. Conclusions

In summary, this investigation has unveiled the presence of the two major lineages of DENV-1 genotype V in Araraquara, SP, and described two clade replacements within this genotype during 2015–2022. These lineages exhibit a history of propagation across various regions of the country before emerging in the municipality. The implications are suggestive that these lineages might have influenced the divergent case counts observed during each epidemic occurrence and played a critical role in the maintenance of DENV-1 circulation. These results emphasize the importance of genomic surveillance as an additional tool in understanding the relationship between viral lineages and the magnitude of outbreaks. Furthermore, our findings have yielded novel insights into the origins of DENV-1 genotype V—a strain of utmost significance within the Brazilian context.

## Figures and Tables

**Figure 1 viruses-16-00274-f001:**
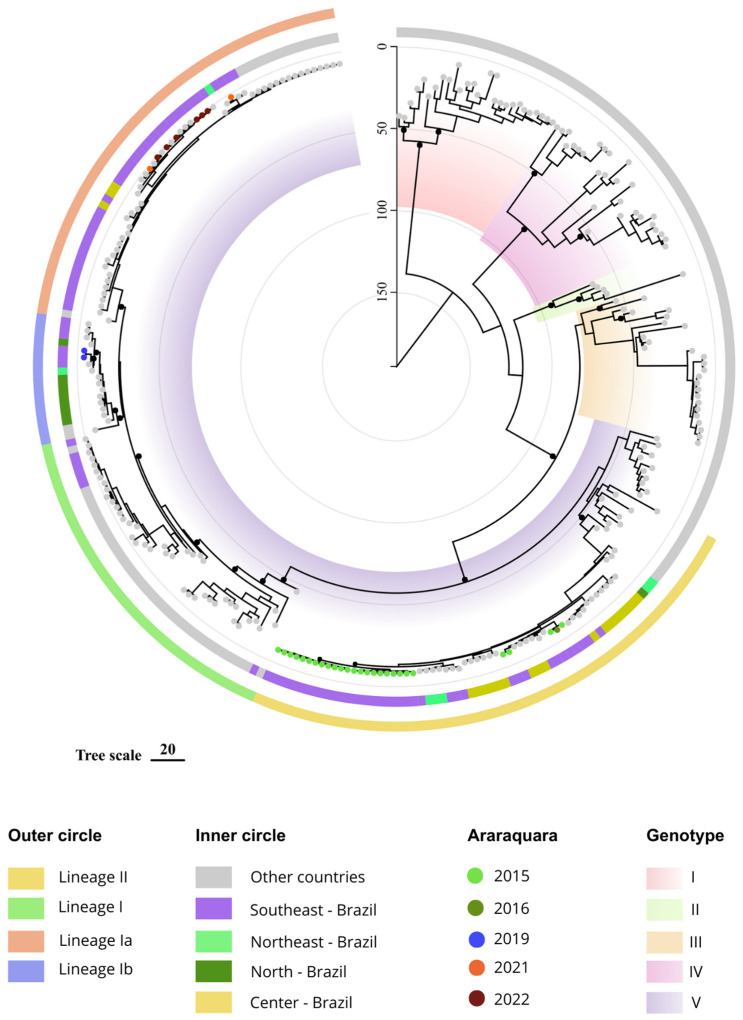
Maximum clade credibility tree of DENV-1. The genotypes I to V are identified by colorful internal blocks. Genotype V, which includes most of the reference sequences and all Brazilian DENV-1, is also classified into lineages and sub-lineages. Small black dots denote the most important nodes with posterior probability >0.7. The tree is scaled by years, represented by vertical line denoting the present (time zero) to 150 years before the present. Sequences from Araraquara were colored according to the year of sampling, 2015–2022 (colorful circles).

**Table 1 viruses-16-00274-t001:** DENV-1 and genotypes’ tMRCAs estimated under the Bayesian Skyline Coalescent and Exponential growth tree priors. The tMRCA and 95% High-Probability Densities are shown for both models, genotypes and lineages. The tMRCA is presented in years.

Group	tMRCA Bayesian Skyline (HPD 95%)	tMRCA Exponential Growth (HPD 95%)
DENV-1	1884 (1833–1920)	1886 (1842–1916)
Genotype I	1966 (1953–1974)	1967 (1954–1976)
Genotype II	1931 (1907–1948)	1930 (1909–1947)
Genotype III	1945 (1927–1958)	1943 (1926–1955)
Genotype IV	1944 (1921–1959)	1944 (1923–1959)
Genotype V	1965 (1955–1972)	1966 (1957–1973)
Lineage I	1977 (1971–1978)	1976 (1973–1978)
Lineage Ia	2001 (1998–2005)	2002 (1999–2005)
Lineage Ib	2001 (1998–2005)	2002 (2000–2005)
Lineage II	1982 (1977–1985)	1976 (1968–1982)
Araraquara 2015	2009 (2006–2011)	2008 (2007–2011)

## Data Availability

Data are contained within the article and Appendix A.

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
