# Peer review of "Phylogenetics, Epidemiology and Temporal Patterns of Dengue Virus in Araraquara, São Paulo State"

_viruses, 2024, doi:10.3390/v16020274_

Round 1
Reviewer 1 Report
Comments and Suggestions for Authors
The manuscript of Souza et al. discusses the dynamics of the Dengue virus (DENV) in Araraquara, Brazil, focusing on DENV-1. Despite yearly epidemics following a pattern of serotype replacement, particularly every three to four years, there is insufficient understanding of DENV-1 circulation in the region. The study generated 37 complete or partial DENV-1 genomes from 2015 to 2022 in Araraquara and reconstructed the evolutionary history using Brazilian and worldwide DENV-1 sequences. The analysis revealed at least three introductions of genotype V in the last ten years, with the time to a most recent common ancestor (tMRCA) for the first sampled lineage estimated at approximately 15 years ago (in 2008). The findings challenge existing assumptions about the emergence time of DENV-1 genotypes, suggesting that genotype V may have diverged more recently than previously thought. The presence of two lineages of genotype V in the municipality may have contributed to the extended persistence of DENV-1 in the region. The study is important and can be accepted in Viruses after minor revisions.
Minor revisions
1) Page 1, lines 37 and 41: update the new classification of ICTV to DENV: Family: Flaviviridae, Genus: Orthoflavivirus, virus species: Orthoflavivirus denguei and virus name: Dengue virus according to https://ictv.global/report/chapter/flaviviridae/flaviviridae/orthoflavivirus.
2) Lines 50 and 51: We also have DENV-3 circulating in Brazil as recently detected in Sao Paulo (https://doi.org/10.1186/s13071-023-06043-1).
3) Other important studies that may appear in the introduction;
a. Souza UJB, Macedo YDSM, Santos RND, Cardoso FDP, Galvão JD, Gabev EE, Franco AC, Roehe PM, Spilki FR, Campos FS. Circulation of Dengue Virus Serotype 1 Genotype V and Dengue Virus Serotype 2 Genotype III in Tocantins State, Northern Brazil, 2021-2022. Viruses. 2023 Oct 24;15(11):2136. doi: 10.3390/v15112136.
b. Gularte JS, Sacchetto L, Demoliner M, Girardi V, da Silva MS, Filippi M, Pereira VMAG, Hansen AW, da Silva LL, Fleck JD, de Almeida PR, Nogueira ML, Spilki FR. DENV-1 genotype V linked to the 2022 dengue epidemic in Southern Brazil. J Clin Virol. 2023 Nov;168:105599. doi: 10.1016/j.jcv.2023.105599.
c. Fritsch H, Moreno K, Lima IAB, Santos CS, Costa BGG, de Almeida BL, Dos Santos RA, Francisco MVLO, Sampaio MPS, de Lima MM, Pereira FM, Fonseca V, Tosta S, Xavier J, de Oliveira C, Adelino T, de Mello ALES, Gräf T, Alcantara LCJ, Giovanetti M, de Siqueira IC. Phylogenetic Reconstructions Reveal the Circulation of a Novel Dengue Virus-1V Clade and the Persistence of a Dengue Virus-2 III Genotype in Northeast Brazil. Viruses. 2023 Apr 28;15(5):1073. doi: 10.3390/v15051073.
d. Amorim MT, Hernández LHA, Naveca FG, Essashika Prazeres IT, Wanzeller ALM, Silva EVPD, Casseb LMN, Silva FSD, da Silva SP, Nunes BTD, Cruz ACR. Emergence of a New Strain of DENV-2 in South America: Introduction of the Cosmopolitan Genotype through the Brazilian-Peruvian Border. Trop Med Infect Dis. 2023 Jun 17;8(6):325. doi: 10.3390/tropicalmed8060325.
e. Giovanetti M, Pereira LA, Santiago GA, Fonseca V, Mendoza MPG, de Oliveira C, de Moraes L, Xavier J, Tosta S, Fristch H, de Castro Barbosa E, Rodrigues ES, Figueroa-Romero D, Padilla-Rojas C, Cáceres-Rey O, Mendonça AF, de Bruycker Nogueira F, Venancio da Cunha R, de Filippis AMB, Freitas C, Peterka CRL, de Albuquerque CFC, Franco L, Méndez Rico JA, Muñoz-Jordán JL, Lemes da Silva V, Alcantara LCJ. Emergence of Dengue Virus Serotype 2 Cosmopolitan Genotype, Brazil. Emerg Infect Dis. 2022 Aug;28(8):1725-1727. doi: 10.3201/eid2808.220550.
4) Page 5, line 181: Remove “but”.
5) Authors must deposit the sequences in GenBank and add the accession numbers to the article.
6) The authors must describe the study's limitations, such as the number of samples, as they are samples from only one municipality in the state of São Paulo, making it difficult to extrapolate the results to other municipalities or Brazilian states.
7) The biggest criticism of the study is the lack of a spatiotemporal diffusion analysis to demonstrate the spread of the virus in the state of São Paulo and other Brazilian states.
8) There was also a lack of detailed discussion comparing the results obtained with recent studies such as Giovanetti et al., 2002; Fritsch et al., 2023; Gularte et al., 2023; and Souza et al., 2023 described above.
Comments on the Quality of English LanguageSmall editing and layout errors that must be corrected by the authors.
Author Response
Dear Reviewer
First of all, we would like to thank you for the important comments and suggestions to improve the quality of our article. In this letter, we presente the responses to each comment and also describe what were the changes in the manuscript, according to your recommendations :
Comments/suggestions
1- We updated the dengue virus classification, following the ICTV criteria (lines 41-44)
2- We updated the text, citing the detection of DENV-3 in the country (line 68)
3; 8- Regarding the comment # 3 and #8: We used the suggested references not only to improve the introduction, but also to make our discussion deeper (lines 80-83, 313-314 and 338-341).
- We thank the reviewer's observation, but to our understanding, the word ¨but¨ is not misplaced, since the branch that emerged in 2009 include all viruses from that year, except those 5 Araraquara viruses. They clustered within the same lineage, but out of the most terminal branch, that includes all the remaining Araraquara sequences.
5- The reviewer raised a very good point. The accession number of the genomes sequenced in this article are now included in the results (lines 250-252).
6- The reviewer is completely correct. The study has limitations, which we tried to address in the revised version of the manuscript. Regarding the comment on the applicability of this information to other municipalities, we acknowledge that extrapolation is impossible. However, we intend to specifically investigate genetic variability and lineage distribution in the municipality of Araraquara. We also recognize that while the results may be restricted to this municipality, the depiction of lineage replacement within a DENV-1 genotype over the years is relevant for a broader understanding of DENV.
7- We understand the reviewer's concern. Indeed, while our work lacks a phylogeographical analysis, one of the articles suggested by the reviewer (and we thank for that) also illustrates the existence of DENV-1 lineage 1 (L1) in the state of Rio Grande do Sul. This finding potentially supports the notion of this lineage emerging as the predominant one in the country, a point we delve into more thoroughly in the updated version of our manuscript. Notably, this lineage was initially characterized by Bruycker-Nogueira, who provided evidence of its sustained presence in the Brazilian territory. Finally, we added this piece of information in lines 338-341.
8- In order to meet the minimal number of words requested for this type of manuscript of the Viruses J., we updated some relevant information in the introduction section (lines 50-64)
Reviewer 2 Report
Comments and Suggestions for Authors
Results - Figure 1 - as this is a manuscript comparing the serotypes, genotypes & lineages of dengue virus from a particular region/state then to have other countries and states in Brazil displayed in the graphic is nice as an overview but impossible for the reader/reviewer to figure out specifically for Araraquara. I propose you have 2 representations, a simplified graphic for everything (as in Fig 1) and a second with just the strains under study.
Comments on the Quality of English LanguageThe manuscript is reasonably well written and requires minor edits as per journal style
Author Response
Dear Reviewer
1- We appreciate the reviewer's suggestion to request more details and clarity in the description of Araraquara samples. However, due to space limitations, we opted to include only one, but very comprehensive figure that could address all the key points of interest from the analyses and results. Moreover, the primary outcome of our work comes from the phylogenetic placement of Araraquara samples in relation to other samples, considering the lineage, the year of sampling, and their relationship with samples collected in other states or countries. Therefore, an additional figure solely for Araraquara would offer limited information in terms of results. In this context, we have included a table with relevant coalescence information from the main branches, emphasizing the crucial Araraquara branch, particularly concerning the 2015 largest cluster. However, we acknowledge that Figure 1 contains a wealth of information, and the legend may not have been sufficiently clear. So, we have added more details to the figure legend to facilitate the identification of the Araraquara samples.
2- In order to meet the minimal number of words of the Viruses J., we updated some information in the introduction section (lines 50-64)
Reviewer 3 Report
Comments and Suggestions for Authors
Manuscript Number: viruses-2762839
Title: "Phylogenetics, Epidemiology and Temporal Patterns of Dengue Virus in Araraquara, São Paulo State"
The article investigates the dynamics of Dengue virus (DENV) circulation, with a focus on DENV-1, in Araraquara, Brazil. The study sequenced 37 complete or partial DENV-1 genomes from 2015 to 2022 in Araraquara, reconstructing the evolutionary history and estimating the time to the most recent common ancestor (tMRCA) for DENV-1, genotype V, and its main lineages. However, as a non-specialist in the field, I found it challenging to fully assess the validity and appropriateness of the methodology employed. Additionally, while the main conclusions regarding the challenges to existing assumptions about the emergence time of DENV-1 genotypes are intriguing, as a non-expert, I am unable to assess the significance of these findings within the broader context of current dengue research.
The introduction section could benefit from a more comprehensive overview of the differences between DENV serotypes, genotypes, and lineages. Clarifying these distinctions early on would assist non-expert readers in better grasping the subsequent content of the manuscript. Moreover, an introduction to the basic structure and characteristics of the DENV genome would be beneficial. Providing a brief but informative overview of the genomic composition would enhance the reader's understanding of the subsequent genetic analyses conducted in the study. Additionally, the Introduction should include a section addressing the early assumptions about the emergence time of DENV genotypes. Introducing these assumptions would contextualize the significance of the study's findings, especially for readers unfamiliar with the historical perspective of DENV research.
It is essential that the authors provide the specific accession codes or identifiers for the sequences they have deposited in public databases. Including these codes is standard practice and is crucial for readers, researchers, and reviewers who wish to access the raw data for further analysis or validation of the study's findings.
A section explicitly outlining the contributions of each author should be included. It is important to accurately reflect the unique contributions of each author to the overall work.
Minor comments:
- The DENV abbreviation is not defined in the abstract. The E abbreviation (line 134) is not defined previously.
- Abstract (lines 23-24) “we generated 37 complete or partial DENV-1 genomes sampled from 2015 to 2022 in Araraquara”. Please change “generated” by another verb, such as “sequenced”.
- The numbers of the nine first references in the References section are not integers (for example 1.0 must be 1)
Author Response
Dear reviewer
First of all, we would like to thank you for the important comments and suggestions to improve the quality of our article.
1- We thank the reviewer comments on the introduction, and we agree that the requested information can be very useful to our readers. Therefore, we added more information in the lines 41-55 and 57-63.
2- The reviewer is absolutely right about the lack of the accession number information. To fix this, we added the ID numbers in the results section, lines 250-252.
3- Thank you, we found this suggestion very important. In fact, this information was requested during the manuscript submission (in the journal form) and may be added in the final version. Below, the reviewer can find what was filled out in the journal form: Conceptualization, Caio Souza and Camila Romano; Methodology, Giovana Caleiro, Ingra M Claro, Jaqueline Goes De Jesus, Thaís Coletti, Camila Da Silva, Alvina Felix, Anderson Vicente De Paula, Expedito Luna, Ester Sabino and Camila Romano; Formal analysis, Caio Souza and Camila Romano; Investigation, Giovana Caleiro, Ingra M Claro, Jaqueline Goes De Jesus, Thaís Coletti, Camila Da Silva, Ângela Costa, Marta Inemami, Andreia Ribeiro, Alvina Felix, Anderson Vicente De Paula, Walter Figueiredo, Expedito Luna, Ester Sabino and Camila Romano; Resources, Ester Sabino and Camila Romano; Writing – original draft, Caio Souza and Camila Romano; Writing – review & editing, Caio Souza and Camila Romano; Supervision, Camila Romano.
4- The DENV abbreviation is now described in the abstract.
5- We thank the reviewer for the suggestion and updated the verb in the line 24 of the abstract.
6- We would like to thank the reviewer for the important suggestion regarding the references and we updated the numbers correctly.
7- In order to meet the minimal number of words of the periodic, we updated some information in the introduction section (lines 50-64)
Round 2
Reviewer 3 Report
Comments and Suggestions for Authors
The authors have improved the article following my suggestions.